# Results of Laboratory Studies of the Automated Sorting System for Root and Onion Crops

**Alexey Dorokhov, Alexander Aksenov, Alexey Sibirev \*, Nikolay Sazonov, Maxim Mosyakov and Maria Godyaeva**

FSBSI "Federal Scientific Agronomic and Engineering Center VIM", 109428 Moscow, Russia; 1053vim@mail.ru (A.D.); alexandr-aksenov@mail.ru (A.A.); sazonov_nikolay@mail.ru (N.S.); vim@yandex.ru (M.M.); airrune@yandex.ru (M.G.)
* Correspondence: sibirev2011@yandex.ru; Tel.: +7-964-584-3518

**Abstract:** The roller and sieve machines most commonly used in Russia for the post-harvest processing of root and tuber crops and onions have a number of disadvantages, the main one being a decrease in the quality of sorting due to the contamination of working bodies, which increases the quantity of losses during sorting and storage. To obtain high-quality competitive production, it is necessary to combine a number of technological operations during the sorting process, such as dividing the material into classes and fractions by quality and size, as well as identifying and removing damaged products. In order to improve the quality of sorting of root tubers and onions by size, it is necessary to ensure the development of an automatic control system for operating and technological parameters, the use of which will eliminate manual sorting on bulkhead tables in post-harvest processing. To fulfill these conditions, the developed automatic control system must have the ability to identify the material on the sorting surface, taking into account external damage and ensuring the automatic removal of impurities. In this study, the highest sorting accuracy of tubers (of more than 91%) was achieved with a forward speed of 1.2 m/s for the conveyor of the sorting table, with damage to 2.2% of the tubers, which meets the agrotechnical requirements for post-harvest processing. This feature distinguishes the developed device from similar ones.

**Keywords:** automation; experiment; roots; sorting; technical vision

## 1. Introduction

Although China has the largest planting area and highest yield of potatoes in the world, the mechanization level of potato production cannot meet the actual demand [1].

Harvesting is the most crucial step in the entire potato production mechanization process [2], and the most challenging issue during this process is to separate potatoes into fractions [3], as well as separating soil and impurities thoroughly and meanwhile controlling the damage rate and bruising rate [4]. Usually potatoes need to be harvested in a short time [5]. Two-stage harvesting involves manual picking, which requires intense labor, yet has a low harvesting efficiency [6]. In order to improve the potato-soil separation performance and to obtain better clod crushing performance [7], most studies have focused on the vibration and arrangement of the separation device [8].

Regarding the bruising and damage control of the potato harvest, the harvesting efficiency and quality can often be improved by developing new types of potato-soil separation devices [9], regulating the operating parameters of the harvester [10], controlling the proportion of potatoes–soil in the potato-soil mixture and enhancing the seedling killing operation [11].

Aiming to meet the demand of heavy soil harvesting conditions in northern China, Li designed a separation device with a two-grade lift chain, in which the length of the two-grade lift chain was 3.1 m [12], the forward speed of the machine was 1.2 m/s and the speed of the lift chain line was 1.5 m/s, the exposed potato rate was 98.1% and the damage rate was 1.1% [13]. In order to improve the effectiveness of separation in a combined potato

harvester, a vertical circular separation transport device was designed, with an average damage rate of 1.46% and an impurity rate of 2.57% [14]. A.B. Kalinin investigated the impact of the coefficient of restitution that is caused by the fall height on potato damage [15], as well as the impact of the contact material and the water content of the potatoes, and identified the main factors affecting the damage to potatoes.

The objectives of all the above studies were to reduce the quality losses in agricultural products by decreasing the number and intensity of impacts in the potato harvesting process. Although there have been a number of scientific studies involving the acquisition of impact information with impact detection sensors, studies exploring the dynamic characteristics of the potato-soil mixture with varying vibration parameters are rare. Therefore, fundamental research on the mechanisms of potato damage is an emerging area.

Impact recording technology and a high-speed camera were used to provide technical recordings to further investigate the separation mechanism and mathematical relationship between maximum exposure acceleration and factors of interest, including tuber feed, grading table conveyor forward speed and grading actuator response time.

Due to the large quantity of the potato-soil mixture and its strong randomness, the separation performance and harvest quality can be easily affected by multiple factors that are related to agricultural machinery and agronomy; hence, it is difficult to obtain high-efficiency separation and a low-loss harvest simultaneously [16].

Potato collisions usually take place many times on separation sieves during the harvesting process [17], where a high rate of rupture, damage and bruising can occur, leading to great economic loss for potato growers. Therefore, a fundamental research on the mechanisms of potato damage is an emerging area [18].

However, it is extremely difficult to carry out a systematic experiment based on existing potato harvesters to acquire ideal results due to the complexity of the system and the high number of variables. For this reason, the work on optimizing and improving the structure of potato harvesters and reducing losses during the mechanized harvesting process are greatly limited. A new separation device for potato separation for a potato harvester with a two-vibration intensity adjusting device has been developed [19]. The objectives of this preliminary study are as follows. Potato-soil separation experiments under different vibration intensity conditions were conducted to improve the harvesting quality.

Response surface analysis experiments were conducted to achieve the desired separation performance.

The presence of soil and plant impurities in the storage heap is one of the most important quality indicators that determines the duration of storage of root crops and onions in the mechanized production of root and onion crops [20].

Currently, the achievement of the specified agrotechnical requirements is ensured only under the maximum permissible operating modes of the clod-breaking and sifting/separating devices in machines made for harvesting root, tuber and onion crops in order to destroy non-passable soil clods, which leads to increased damage and the loss of separated products. This circumstance is due to delays and gaps in the development of the technological foundations, technologies and functioning elements which would make it possible to reduce or eliminate the content of mechanical impurities in the harvesting of root and tuber crops and onions.

Post-harvest and pre-planting onion treatments provide for the separation of soil and plant impurities, as well as the isolation of small and substandard bulbs and sorting. These operations are very important in creating favorable conditions during storage with natural and active ventilation of the bulbs, and also affect the reduction of total onion losses during storage, improving the quality of the sorting process and the commercial preparation of the bulbs.

Impurities such as soil and plant residues are removed using special mechanisms, which are based on various designs.

Therefore, the efficiency of the separation process is largely determined by the types, moisture and physical and mechanical properties of impurities, including soil lumps and plant residues, supplied with the bulbs.

Currently, several basic technological methods of separating impurities from bulbs are used. Their effectiveness is characterized by assessing the damage to the bulbs, the productivity of the devices and the completeness of the separation of the corresponding types of impurities.

Despite this, comparative studies of various sorting surfaces show that active working bodies (rollers, disks, springs) have significantly higher (up to 30%) specific productivity than devices based on a different principle of operation.

Roller and sieve machines for the post-harvest processing of root and tuber crops and onions, which are the most common ones used in Russia, have a number of disadvantages. The main disadvantage is a decrease in the quality of sorting due to contamination of working bodies, which increases the amount of losses during sorting and storage.

To obtain high-quality competitive production, it is necessary to combine a number of technological operations during the sorting process, such as dividing the material into classes and fractions based on quality and size, as well as identifying and removing damaged products.

In order to improve the quality of sorting of root tubers and onions by size, it is necessary to ensure the development of an automatic control system to meet operating and technological requirements, the use of which will eliminate manual sorting on bulkhead tables in post-harvest processing.

To fulfill these conditions, it is necessary to develop an automatic control system with the ability to identify the material on the sorting surface, taking into account its appearance, damage and the automatic removal of impurities. This feature distinguishes the developed device from similar ones.

## 2. Materials and Methods

The sorting line for roots, onions and tuber crops with an automatic control system for the given operational and technological parameters consists of a receiving hopper connected to transfer conveyor, a spiral heap cleaner and a conveyor sorting device with two modular conveyor belts (Figure 1).

According to this design, the conveyor sorting device has a technical vision system: a camera with a coverage area of the entire working surface of the two conveyor belts, the control unit and the actuators with elastic-elastic working bodies, driven by electric drives.

To recognize substandard commercial products, a multipurpose web camera "Logitech HD Pro C920" (2 mpx matrix, 1920 × 1080 (Full HD), Carl Zeiss optics, USB connected) is installed over the conveyor belts (labeled 5 in Figure 1), which has a high resolution and high detail. A schematic block diagram of the operation of the electronic line system is presented in Figure 2.

To determine the optimal technological parameters of the machine for the post-harvest processing of root and tuber crops and onions with an automated sorting system, experimental studies were carried out in a laboratory setup, the general structure of which is shown in Figure 3.

The line for the automated post-harvest sorting of root crops and potatoes consists of a receiving hopper connected to a transfer conveyor, spiral heap cleaner and conveyor sorting device, with two modular conveyor belts. The conveyor sorting device has a vision system in the form of a camera with a coverage area of the entire working surface of the two conveyor belts, a control unit and executive mechanisms with elastic-elastic working bodies, driven by electric drives. There are trays for the removal of the affected and damaged root crops, as well as trays for sorting quality products based on their size. Electric motors drive the transfer conveyors, spiral heap cleaners and the conveyor sorting device.

The receiving hopper, transfer conveyor and heap cleaners are designed to receive processed products and prepare them for sorting by size.

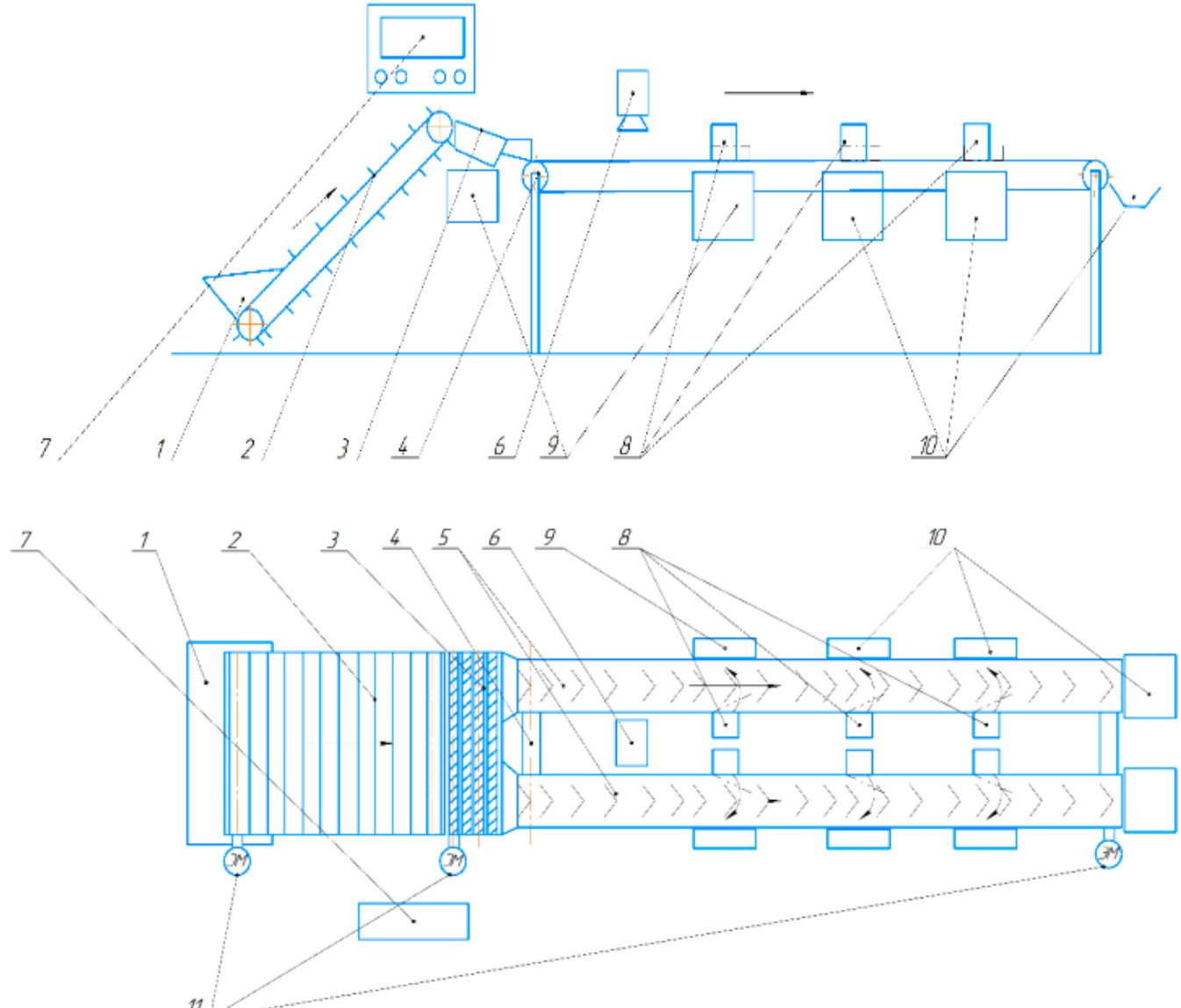

**Figure 1.** Technological scheme of a machine for the post-harvest treatment of potato tubers with an automated sorting system: 1—receiving hopper; 2—transfer conveyor; 3—spiral heap cleaner guides; 4—conveyor sorting device; 5—conveyor belts; 6—camera; 7—control unit; 8—executive mechanisms; 9 and 10—trays; 11—electric motors.

A conveyor sorting device with two conveyor belts serves to separate root and tuber crops. The conveyor belt has a directional herringbone-type relief, which contributes to a more detailed reading of the main linear dimensions via the technical vision camera and the precise movement of root and tuber crops into the trays using actuators with elastic-elastic working bodies.

The actuator, in the form of elastic-elastic working bodies, is selected in such a way as to exclude damage to the surface of root and tuber crops, but at the same time creating sufficient force to move them into trays.

The combination of operations will reduce the duration of the technological process and the number of workers, reduce the damage rate of potato tubers, reduce the number of post-harvest processing operations, preserve the commercial qualities of root crops and bulbs, reduce production costs and the cost of grown products, increase the yield of cultivated crops and, ultimately, reduce energy costs and production costs.

The structure of a sorting table machine with an automated sorting system is shown in Figure 4.

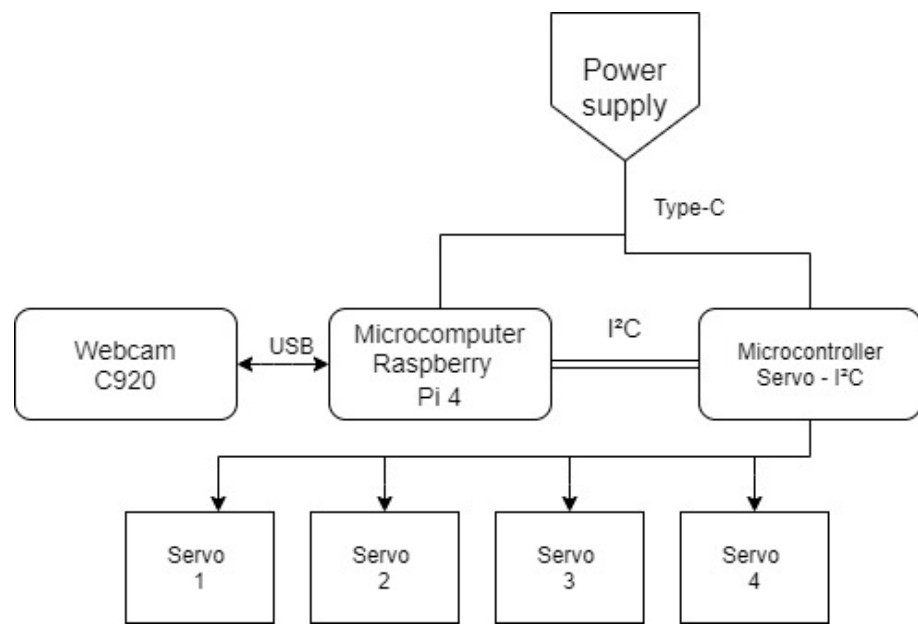

**Figure 2.** Block diagram of an automated sorting line for root, tuber and onion crops.

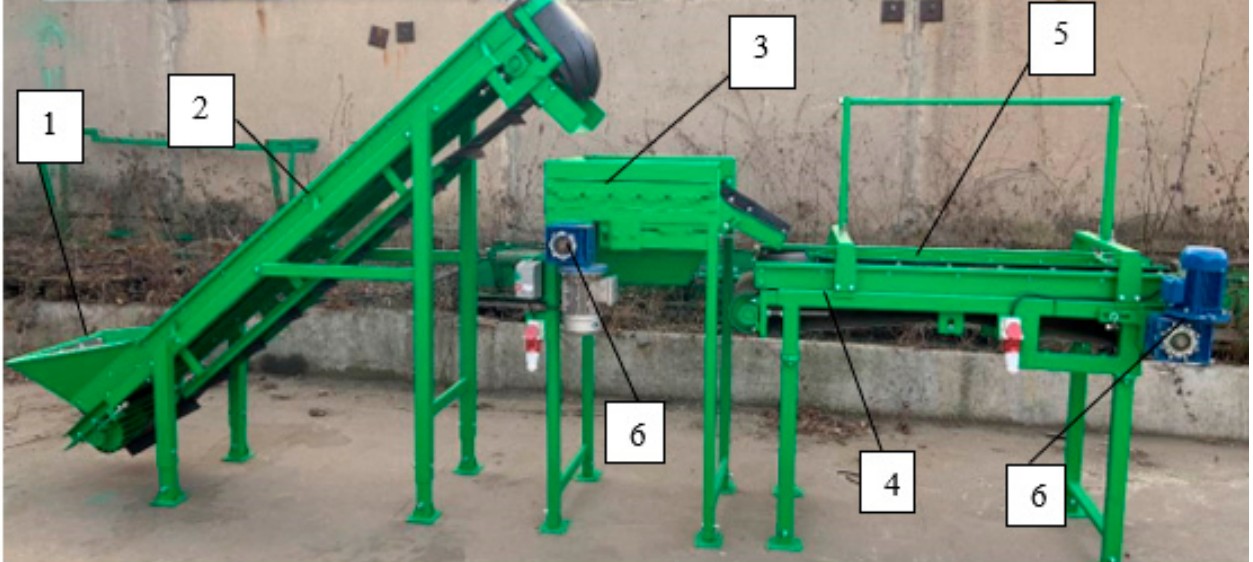

**Figure 3.** Photograph of a machine for the post-harvest processing of potato tubers with an automated sorting system: 1—receiving hopper; 2—transfer conveyor; 3—spiral heap cleaner; 4—sorting table; 5—conveyor belts; 6—electric motors.

An important indicator characterizing the quality of sorting devices in the post-harvest processing of potato tubers is the accuracy of separation into size fractions. When dividing potato tubers based on dimensional characteristics, the quality of the sorting surface is determined by the theoretically possible sorting accuracy, which depends on the potato variety, the size characteristics used for separation, the fractional composition, as well as the design features of the working bodies. Additionally, the accuracy of separation is an important indicator characterizing the quality of the sorting devices during the post-harvest processing of root and tuber crops.

When dividing commercial products according to dimensional characteristics, the quality of the sorting surface is determined by the theoretically possible sorting accuracy, which

depends on the varieties of the root and tuber crops, the size characteristic for separation, the fractional composition and the design features of the working bodies (Figure 5).

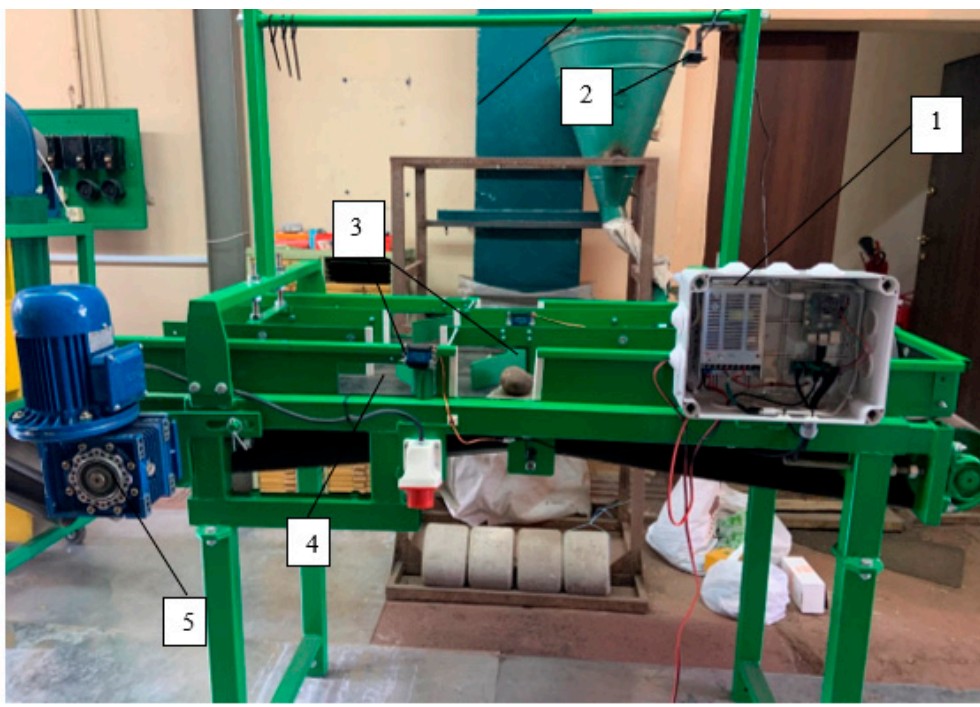

**Figure 4.** Photograph of a sorting table with an automated sorting system: 1—control unit; 2—camera; 3—executive mechanisms; 4—trays; 5—conveyor belt driven by an electric motor.

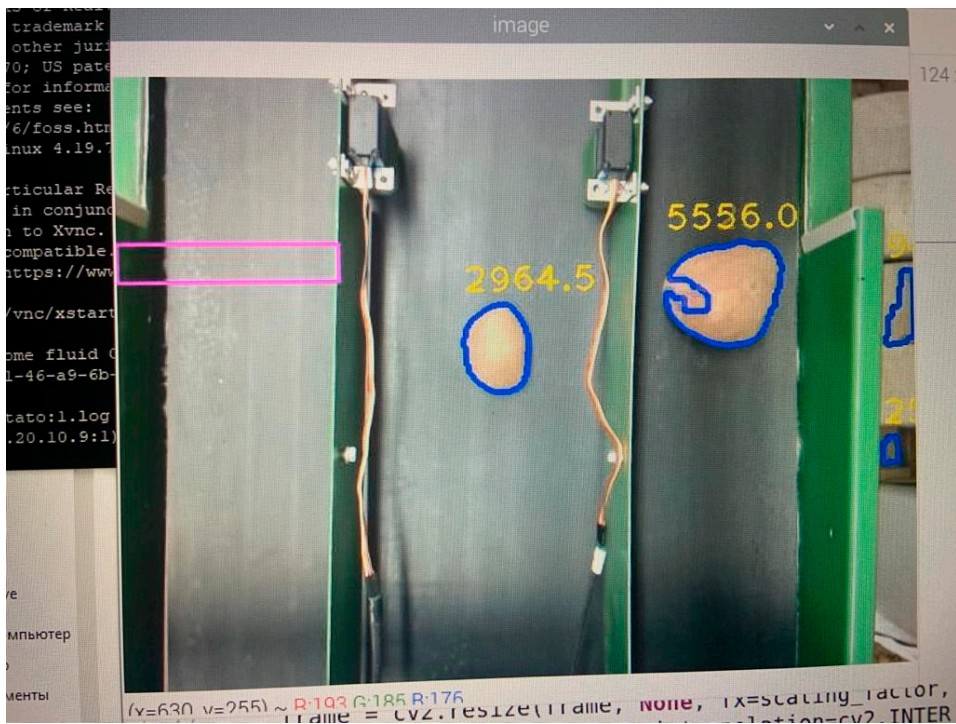

**Figure 5.** General structure of recognition of root and tuber crops and bulbs, as well as their dimensional characteristics.

The machine's theoretically possible screening precision for the accepted dimensions of particular onion varieties may be determined by size/mass or variational diagrams and correlation tables compiled for each variety for a given yield. The power supply of the main software and hardware of the line for the post-harvest processing of root and tuber crops and onions comes from the AC mains. Machine vision systems can detect and classify features that can be quantitatively measured (such as potato size) or approximated by quantitative models (such as shape and external defects) through image processing and analysis.

For the recognition of substandard commercial products above the conveyor belts, a multipurpose web camera (7) "Logitech HD Pro C920" is installed on the bracket (6), which scans objects with high detail. The sorting of potato and onion tubers is carried out based on thickness and diameter by programming the Raspberry Pi4 microcomputer through the creation of a database of the dimensional characteristics of the studied crops. The algorithm and flowchart of potato tuber sorting and image recognition process [21] created for analyzing the obtained spectral images of potato tubers includes the following steps (Figure 6):

1. Loading the spectral image in HDR format (file—open file command).
2. The result is the uploaded image in preview mode (RGB composite colors).
3. Highlighting key points. Key points are:

    a. Dimensional characteristics of the tuber (length and width) without visible damage to substandard products.
    b. Tuber surface areas without visible damage to conditioned products.
    c. Tuber surface with visible signs of damage.

4. As a result, the key points are highlighted and the reflection spectra are extracted from the corresponding pixels of the spectral image (point-based for point-based regions of interest, averaged for extended/figured regions of interest).
5. Analysis of the reflection spectra of the size characteristics of the tuber (length and width) in spreadsheets.

    a. Calculation of the size distribution of the standard deviation and/or coefficients of variation for the data sets that include the size characteristics of tubers:

        i. With different degrees of tuber damage.
        ii. With different quantitative content in terms of the size characteristics of the tuber.

    b. A search for ranges with high and low variation in the size characteristics of the tubers.
    c. Comparison of the calculated size characteristics of the tuber (length and width) with the main characteristics from existing databases.
    d. Selection of spectral bands:

6. Output of the size characteristics of the tuber based on the results of the analysis, entering it into the database of the developed program by the team, and transferring it to the executive mechanism.

This study is aimed at creating approaches to the interpretation of these dimensional characteristics of potato tubers that are characteristic of high-quality sorting and various types of damage.

As a measurement of the diameter of the cross-section (Z), we use the small axis of the ellipse (B), which shows the division of potato tubers by the value $l \times b$, which most fully determines the mass characteristics of the tuber (m) (Figure 7):

$$m = l \times b \tag{1}$$

where l is the length of the tuber m, and b is the width of the tuber m.

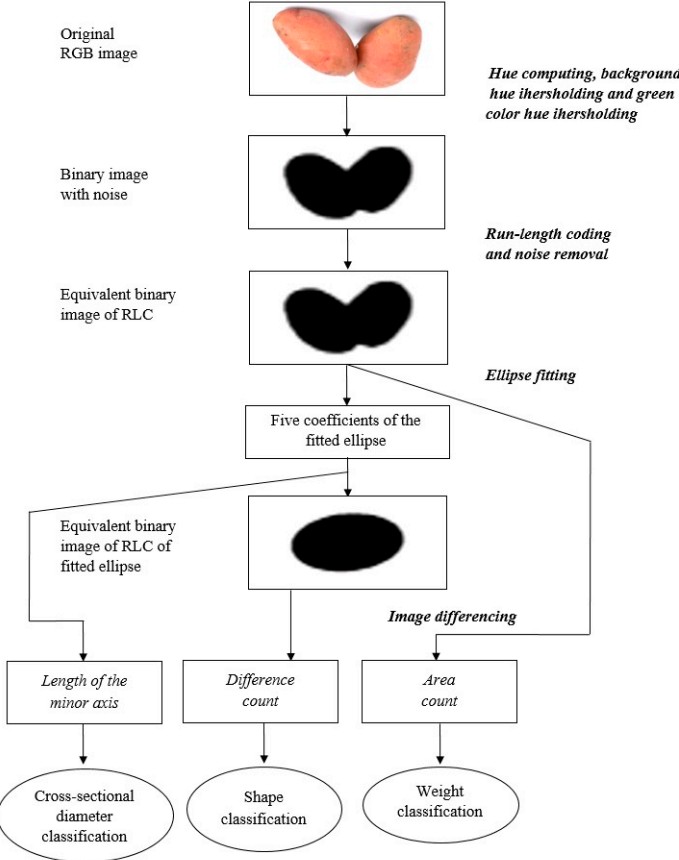

**Figure 6.** Flowchart of potato tuber sorting and image recognition process. The diameter of the cross-section of the potato tuber determines the size requirement when sorting potatoes. It is defined as the largest dimension perpendicular to the longitudinal axis.

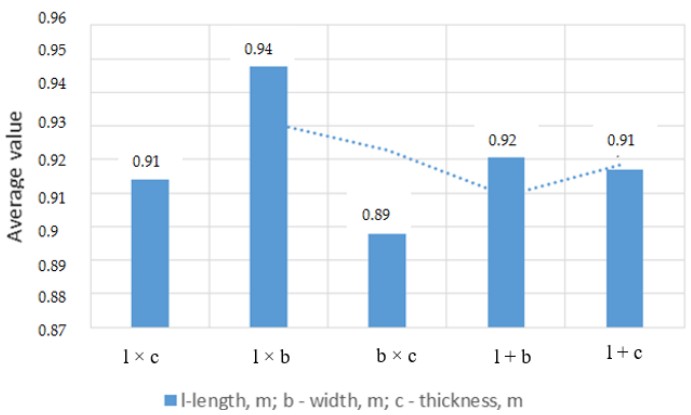

**Figure 7.** A histogram of the correspondence of the separation of potato tubers by values.

To determine the above analytical dependence, experimental studies were conducted to determine the dimensional and mass characteristics of potato tubers of the varieties "Nevsky" and "Red Scarlet" in a 3-fold repetition of 800 measurements, the dispersion diagrams of which are shown in Figure 8.

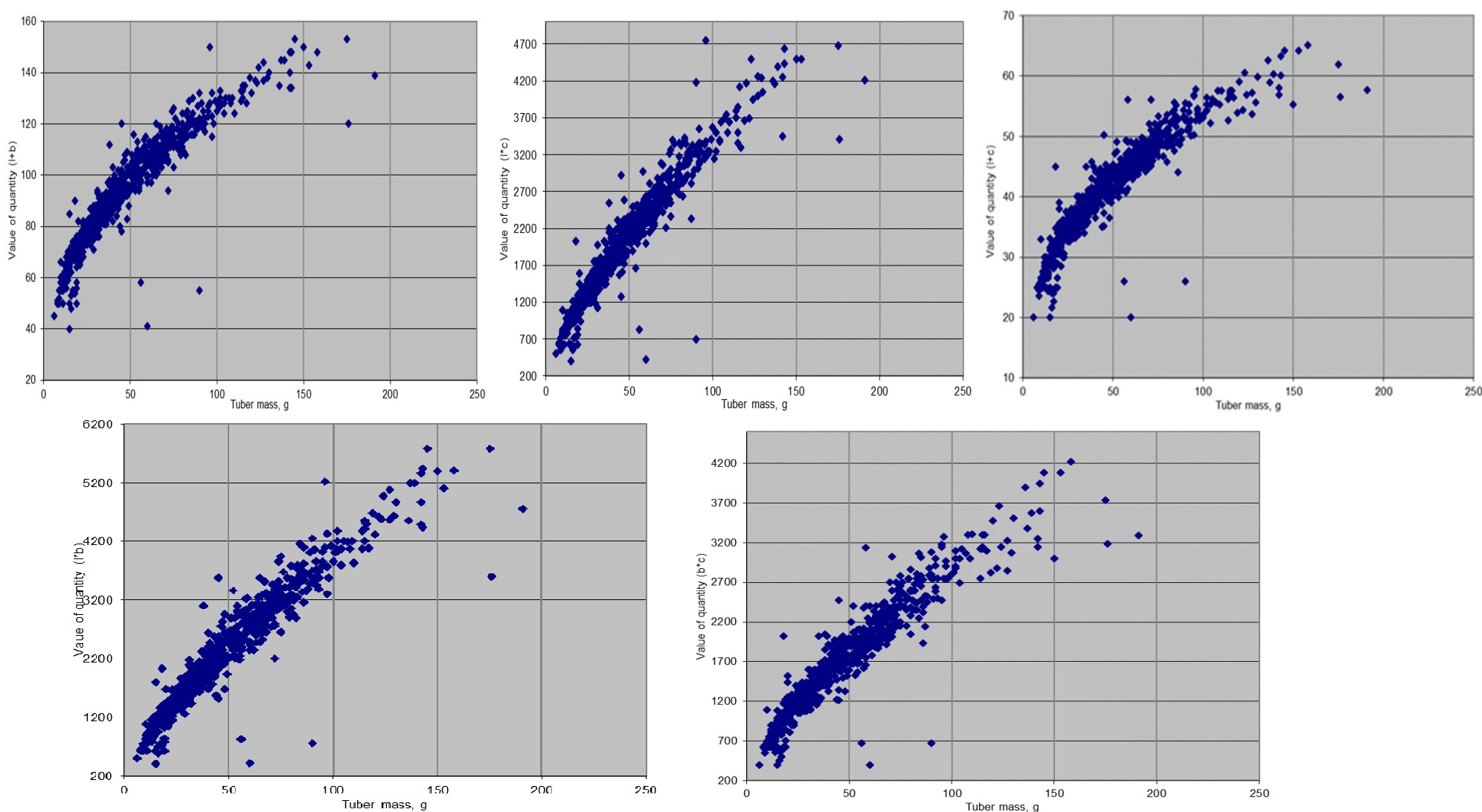

**Figure 8.** Scattering diagram of the results of the studies, determined by the values (l × b, l × c, b × c, l + b, l + c) of the correlations with the mass values.

Clearly, the visual observation of the scattering in the number of stand-out points can be used to assess the level of influence of the studied variables (l × b l × C, b × c, l + b, l + c) that are most closely correlated with the mass of the tuber (m).

The analysis of graphical dependencies indicates that the value of l×b fully confirms the analytically-dependent sortation of potato tubers by weight, depending on the work of its parameters.

Based on this, the main features of the evaluation models for determining the length and width of potato tubers created for "point" measurements were rethought, and approaches for detecting damage to tubers in spectral images and their dimensional characteristics were developed.

A DNaT-400 lamp is used in the light system of the sorting surface to increase the scanning ability of commercial products (Figure 9).

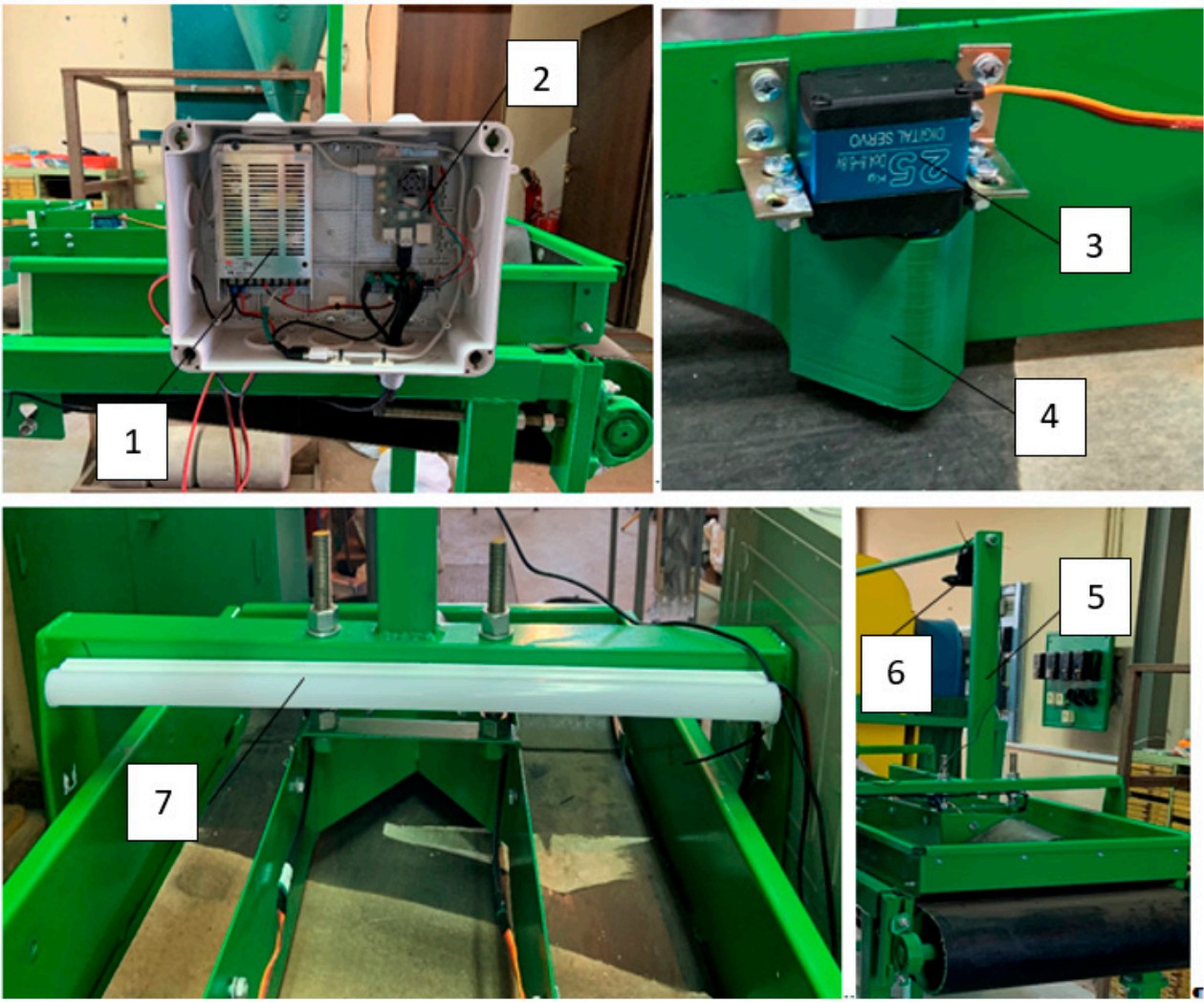

**Figure 9.** Software and hardware of the automatic line control system for the post-harvest processing of root and tuber crops and onions: 1—microcontroller "Arduino Mega 2560"; 2—power supply unit; 3—servo drive "SG90"; 4—actuator; 5—camera mounting bracket; 6—"Logitech HD Pro C920" webcam; 7—DNaT-400 high pressure gas discharge lamp.

It is installed above the working surface of the conveyor belts. The setting of the light mode for a time period, as well as the order of operation of the servos, is carried out using a programmable microcontroller.

To refine the algorithm for recognizing potato tubers, studies of the size/mass and physical-mechanical properties of potato tubers "Nevsky" and "Red Scarlet" were carried out.

Analysis of the size/mass characteristics of potato tubers was carried out on samples of the 2019 harvest, after post-harvest treatment and subsequent warm ventilation storage in a warehouse that met the requirements for storing potatoes.

Sampling was carried out in April 2020 in accordance with GOST 12036-85 "Seeds of agricultural crops. Acceptance rules and sampling methods".

Furthermore, the mass measurement of tubers was carried out on a laboratory scale "Citizen CY" device, which allows results to be obtained with an accuracy of 0.001 g, and the main dimensional characteristics of the bulbs were determined with an accuracy of 0.05 cm.

Each parameter of the size/mass characteristics was measured in triplicate, after which the average values of mass measurements were used to estimate the variation series.

At the same time, the concepts and elements generally accepted in variation statistics to characterize the variation series were used: average variation—X, standard deviation—σ, coefficient of variation—ν. Each of the main elements was determined according to the known formulas of variation statistics.

This made it possible to determine the accuracy of the experimental data and to establish the acceptable limits within which they were sufficiently reliable.

To determine the number of intervals (K) for varying the values of the parameters of the size/mass characteristics of tubers, we used the empirical relationship:

$$K = \sqrt{n}, \tag{2}$$

where n is the number of tubers, pcs.

$K = \sqrt{100} = 10$.

The sampling range can be expressed as:

$$R = xmax - xmin, \tag{3}$$

where xmax and xmin denote the maximum and minimum values of the investigated feature.

The interval of the investigated feature is calculated as follows:

$$D = R/K. \tag{4}$$

The size/mass characteristics of tubers combines the following features: the shape, size, and weight.

While planning a multifactorial experiment, it is necessary to analyze and determine the input parameters of the sorting process which most significantly affect the quality indicators of the sorting. The selected factors should be controllable, unambiguous, compatible, independent, and the accuracy of measurements of the factor levels should be higher than the accuracy of the optimization parameter values.

The accuracy of dividing tubers into fractions, which characterizes the quality of the device and is determined by the coefficient of accuracy of sorting (GOST R 51,808—2001) is used as an optimization factor:

$$K = (mi/m) \times 100, \tag{5}$$

where mi represents the mass of tuber fractions with deviations in quality and size, in kg, and m is the total mass of tubers in the sample, in kg.

Weighing results were recorded up to the second decimal place. Then, the tubers were visually inspected for mechanical damage and spots on the surface, and tubers with no peel on more than half of the surface. Based on the inspection results, the products in each of the samples were assigned to the corresponding fraction, depending on compliance with certain standards.

After choosing the optimization parameters, we took the main factors influencing the sorting process.

The following factors were selected as the studied factors:

- Supply of tubers $Q_A$, kg/s;
- Forward speed of the conveyor of the sorting table $v_{TR}$, kg/s;
- Response time of the sorting actuator $t_A$, s.

To determine the optimal technological parameters of the line for the post-harvest processing of root and tuber crops and onions with an automatic control system, experimental studies were carried out on a laboratory installation, the general view and diagram of which are shown in Figures 1 and 3, respectively.

The $Q_A$ amount of tubers was set by changing the productivity of the feeding conveyor equipped with a receiving hopper, using the control panel, from 4 to 8 kg/s. The supply speed of the conveyor of the sorting table $v_{TR}$ was varied from 0.8 to 1.2 m/s, using a frequency converter from the Tecorp Group. During the experiments, the sorted tubers were collected in separate containers for different fractions. Before starting the experiment according to the described methodology, the determination and analysis of the physical and mechanical properties of tubers was carried out.

For each factor, three levels were selected: lower, upper and main—zero level. After that, the interval of variation of the factors was established (Table 1).

**Table 1.** Levels of variation of factors when sorting bulbs with the line for the post-harvest processing of root and tuber crops and onions with an automatic control system.

| Variation Levels | Variable Factors | | | Optimization Criteria |
|---|---|---|---|---|
| | Supply of Tubers $Q_A$, kg/s | Sorting Table Conveyor Speed $v_{TR}$, kg/s | Response Time of the Sorting Actuator $t_A$, s | Accuracy Factor for Sorting Tubers K, % |
| | | Variation Interval, $\Delta x_i$ | | |
| | 2 | 0.2 | 1 | |
| upper (+1) | 8 | 1.2 | 3 | 85.0 |
| lower (−1) | 4 | 0.8 | 1 | 76.6 |
| basic (0) | 6 | 1.0 | 2 | 80.0 |
| Designations | $X_1$ | $X_2$ | $X_3$ | Y |

After that, to check the compliance of the tubers isolated in this fraction with the requirements for this fraction, their size/mass characteristics were determined. Laboratory investigations were performed in order to determine the optimal technological parameters of the line for the processing of roots and tubers and onions with an automated control system in laboratory conditions.

## 3. Results and Discussion

Finally, after analyzing the results of the multi-factor experiment using the computer program "STATISTICA—10.0", the results of the functional response were collected—the accuracy coefficient when sorting tubers under varied factors, according to the second-degree design of Box–Benkin, an accurate mathematical model describing the dependency of the sorting quality of tubers on the selected factors K = f ($Q_A$,$v_{TR}$,$t_A$) can be expressed as

$$Y = 91.41 + 0.82x_1 - 1.53x_2 + 0.93x_3 - 1.93x_1{}^2 - 2.31x_2{}^2 - 0.96x_3{}^2 - 3.47x_1x_2 - 1.25x_1x_3 - 0.42x_2x_3. \tag{6}$$

The hypothesis regarding the adequacy of the second-degree model was checked by means of the statistical analysis of the regression equation. The results of the calculation of the statistical characteristics are presented in Table 2.

**Table 2.** Statistical characteristics of the errors in the experiment.

| № | $Y_1$ | $Y_2$ | $Y_3$ | $\overline{Y_u}$ | $\hat{Y}_u$ | $S_y^2$ | $S_{LF}^2$ | $(\overline{Y_u}-\hat{Y}_u)^2$ |
|---|-------|-------|-------|------|------|------|-------|------|
| 1 | 82.4 | 84.3 | 83.2 | 83.3 | 84.26 | 0.89 | 0.256 | 0.94 |
| 2 | 82.6 | 86.2 | 83.8 | 84.2 | 85.47 | 0.06 | 0.35 | 1.63 |
| 3 | 91.2 | 90.6 | 92.4 | 91.4 | 92.4 | 0.09 | 0.27 | 1.02 |
| 4 | 89.5 | 90.3 | 89.3 | 89.7 | 91.27 | 0.14 | 0.48 | 2.49 |
| 5 | 89.6 | 90.7 | 90.3 | 90.2 | 91.44 | 0.138 | 0.22 | 1.56 |
| 6 | 85.2 | 84.6 | 83.4 | 84.4 | 85.6 | 0.612 | 0.29 | 1.46 |
| 7 | 89.9 | 89.5 | 89.7 | 89.7 | 92.7 | 0.033 | 0.62 | 9.06 |
| 8 | 90.3 | 88.2 | 90.6 | 89.7 | 90.48 | 0.198 | 0.17 | 0.62 |
| 9 | 86.2 | 85.3 | 85.6 | 85.7 | 87.48 | 0.016 | 0.57 | 3.2 |
| 10 | 88.9 | 89.4 | 90.8 | 89.7 | 90.61 | 0.12 | 0.706 | 0.84 |
| 11 | 86.5 | 84.2 | 86.4 | 85.7 | 87.24 | 0.107 | 0.55 | 2.4 |
| 12 | 91.6 | 90.7 | 91.9 | 91.4 | 92.28 | 0.13 | 0..216 | 0.79 |
| 13 | 89.9 | 92.6 | 91.7 | 91.4 | 91.7 | 0.11 | 0.02 | 0.09 |
| 14 | 92.8 | 89.4 | 92.0 | 91.4 | 93.5 | 0.042 | 0.33 | 4.41 |
| 15 | 91.5 | 92.3 | 90.4 | 91.4 | 93.0 | 0.028 | 0.69 | 2.56 |
| Σ | - | - | - | 1266.4 | - | 2.814 | 5.538 | 33.07 |

The values of Fisher's criterion $F_T$ at the 5% level of importance for the resulting equation at the degree of freedom of the numerator $v1 = No - (F_T + 1) = 11$ and the denominator $v2 = No(m - 1) = 30$ was chosen in the aforementioned table as 2.1.

$$v2 = No(m - 1) = 15\,(3 - 1) = 30 \tag{7}$$

where No denotes the number of experiments and m is the number of factors in the design matrix.

$$F_T = F_T = S_{LF}^2/S_y^2 = 5.538/2.814 = 1.968 \tag{8}$$

The calculated value of the Fisher criteria F = 1.97. Because $F_T$ = 2.1 > F = 1.97, we obtain an adequate result.

Analyzing Figure 10, it can be seen that the accuracy of the sorting of tubers is 91% at the optimal levels of the factors in consideration: the supply of tubers $Q_A$ = 6.5–6.9 kg/s, the time of the start of the sorting mechanism $t_A$ = 1.9–2.3 s, and the forward movement speed of the transporter's sorting table is in the interval between $v_{TR}$ = 0.62–0.75 m/s.

Equation (6), in which we take into consideration the regression value and coefficients, can be presented in the following format:

$$\begin{aligned} Y = &-482.817 + 32.13Q_A + 499.51v_{TR} + 308.95t_A - 0.62Q_A{}^2 - 1536.67v_{TR}{}^2 \\ &- 71.35t_A{}^2 + 9.03Q_Av_{TR} - 4.75Q_At_A - 259.09v_{TR}t_A \end{aligned} \tag{9}$$

The graphical display presented in Figure 10 allows us to determine the levels of variation of the investigated factors, the values of which were confirmed in production conditions. The abscissa and ordinate axes represent the selected factors affecting the sorting accuracy, displayed as color areas.

During the performance of the investigation of production in relation to the sorting lines for roots and tubers and onions with an automated system which controls the regime and technological parameters to find the optimal value of the forward movement speed of the transporter's sorting table $v_{TR}$, all the selected parameters during the laboratory research except for $v_{TR}$, were constant at their optimal levels, found during the laboratory research to be the following: supply of tubers $Q_A$ = 6.5 kg/s, the time of the start of the sorting mechanism, $t_A$ = 1.4 s.

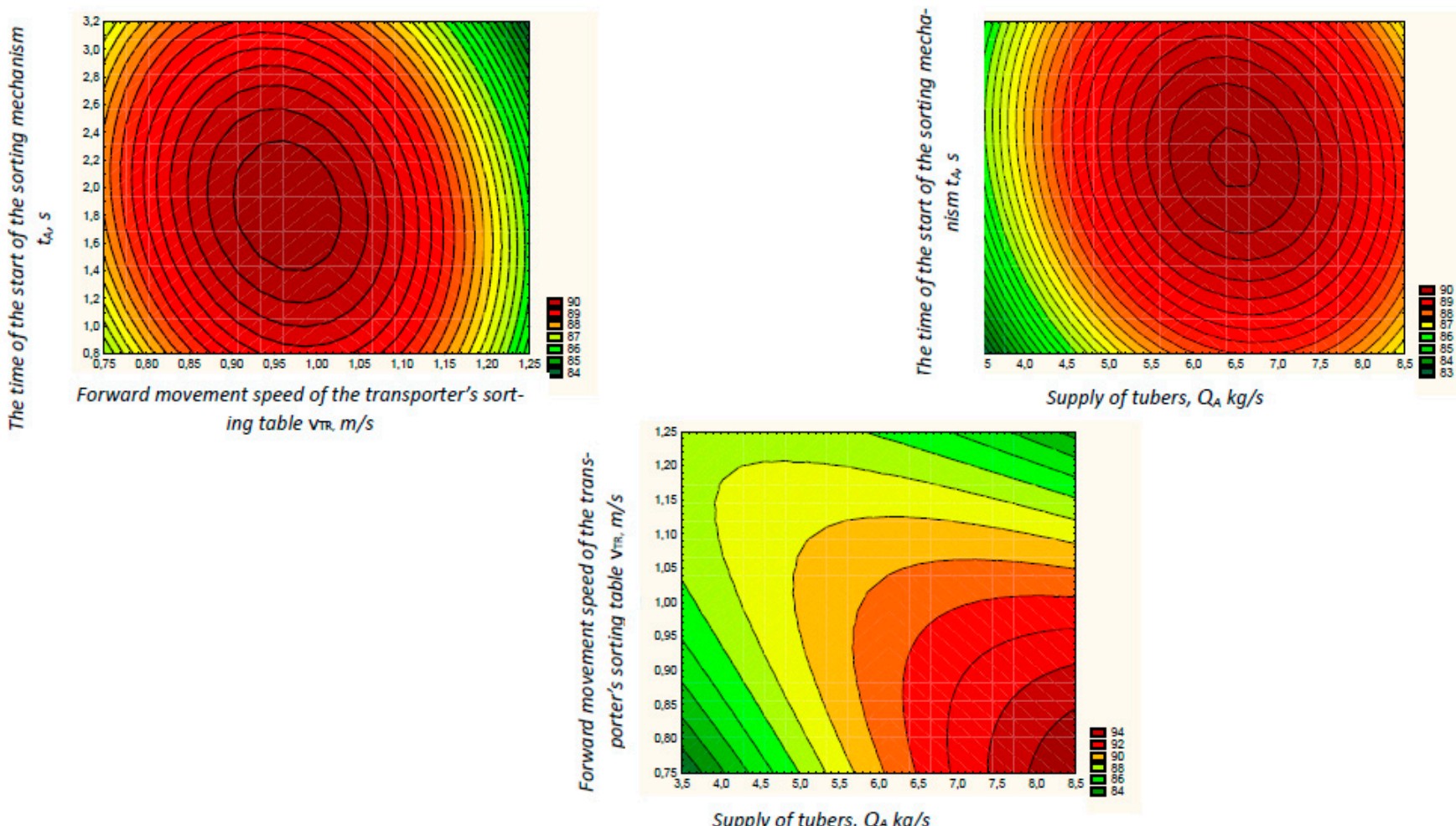

**Figure 10.** Two-dimensional dissection of the response surface, describing the dependence of the accuracy of the sorting of the tubers on the selected factors: forward movement speed of the transporter's sorting table ($v_{TR}$), supply of tubers ($Q_A$) and the time of the start of the sorting mechanism ($t_A$).

As a result of the analysis of the experimental data, we built the following graphs, which show the dependency of sorting precision and damage to bulbs on the speed of the forward movement of the transporter's sorting table (Figure 11).

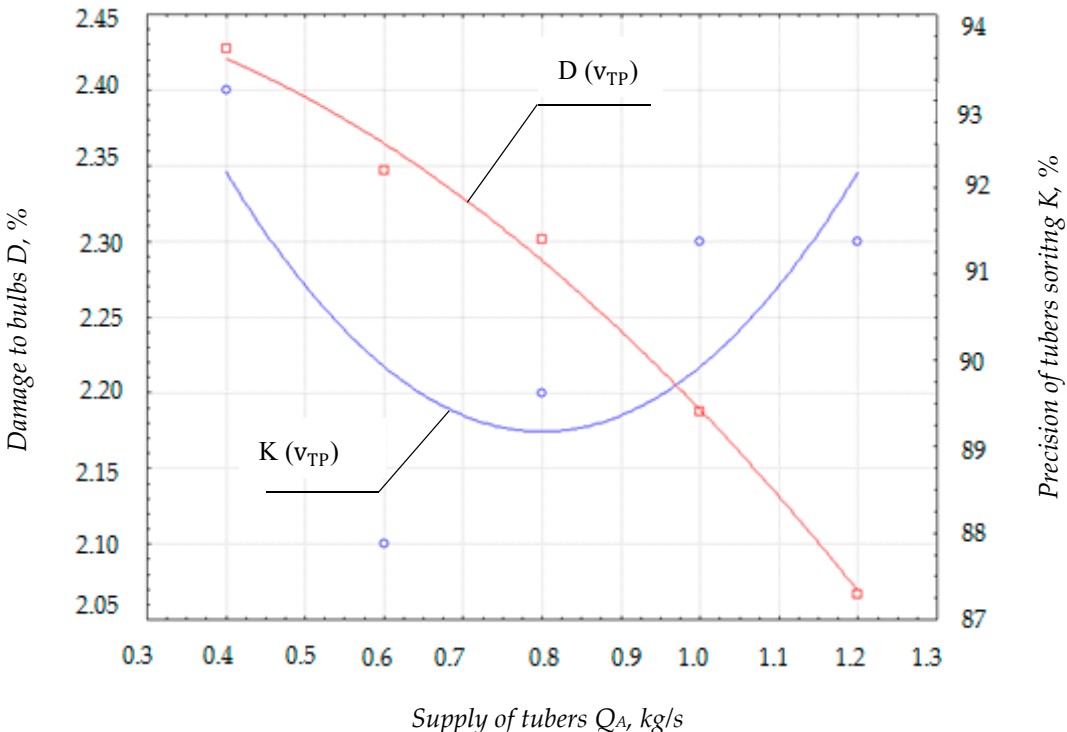

**Figure 11.** Dependency of the precision of the sorting (K%) and damage to the tubers (D%) on the speed of the forward movement of the transporter's sorting table.

The correlation between the precision of the sorting and the damage to the tubers on the speed of the forward movement of the transporter is represented by the calculation of the parabolic functions:

$$D = 2.86 - 1.71v_{TR} + 1.07v_{TR}^2,$$
$$K = 94.34 - 0.27v_{TR} - 4.64v_{TR}^2. \tag{10}$$

After analyzing the graph (Figure 11), we can claim that the maximum accuracy of the sorting of the tubers (>91%) is achieved with the forward movement speed of the transporter's sorting table set at 1.2 m/s, with damage to the tubers at 2.2%, which is in accordance with the agrotechnical norms for post-harvest processing.

As a result of the laboratory analysis of the post-harvest processing line for roots, onions and tubers with an automated control system, the device manages to achieve 90% precision in the sorting of potato tubers as a result of the use of its automated sight system, employing a camera with a field of view covering the entire working surfaces of the two transporter lines, along with the optimal parameters for the research: the supply of tubers, the forward movement speed of the transporter's sorting table and the speed of the processing mechanism in the following limits: $Q_A$ = 8–8.5 kg/s, $v_{TR}$ = 0.62–0.75 m/s and $t_A$ = 1.9–2.3 s.

The obtained results confirm the prospects of the ongoing work on the development of devices for sorting marketable products, since they are consistent with the modern level of technological development for the post-harvest sorting of root tubers, concerning work on the use of machine learning tools, vision and the recognition of marketable products.

Heinemann et al. developed a prototype inspection station to grade potatoes according to the United States Department of Agriculture inspection standards. Their station

consisted of an imaging chamber, a belt conveyor, a camera, a sorting unit and a computer for image acquirement and analysis, as well as for equipment control. Their system managed to accurately classify 80%, 77% and 88% of the moving potatoes in three runs at the speed of three potatoes per minute in online mode, and correctly classified 98%, 97% and 97% of offline stationary potatoes in another three runs [22].

Rios-Cabrera et al. managed to extract potato properties by means of image processing. Their goal was to find out the quality and to evaluate the physical properties of potatoes using artificial neural networks i.e., ANNs. A method which was named blurred ARTMAP because of its stability and convergence speed outperformed the other models. They suggested a few algorithms to determine potato defects like greening, scabs and cracks, which could be efficiently used to grade potatoes of different qualities [23].

The above technologies for sorting potato tubers, a characteristic feature of which is the separation of marketable products into fractions using machine vision and various types of actuating mechanisms, provide an accuracy of separation ranging from 77% to 97.4% when feeding two potatoes per second, which provides an increase in the performance of sorting [22–24] in comparison with the developed system of sorting potato tubers, which provides a maximum sorting accuracy of more than 93% when feeding potato tubers from $Q_A$ = 0.3 kg/s.

## 4. Conclusions

As a result of this study, a block diagram and an algorithm for the process of sorting potato tubers and recognizing images of tubers were developed. Furthermore, it was determined that the most complete separation of potato tubers by weight, depending on the product of its parameters, is satisfied by the value $l \times b$, which most fully determines the mass characteristics of the tubers (m). This is shown in the histogram and the scattering diagram of the results of studies using the correlation values ($l \times b, l \times c, b \times c, l + b, l + c$) with the mass, as shown in Figures 7 and 8, respectively.

A laboratory setup with an automated system was created to determine the adequacy of the developed analytical dependence (1) and to assess the efficiency of the sorting algorithm, as shown in Figures 3 and 4. The results indicated a positive trend in the process of sorting potato tubers into fractions with an accuracy of more than 93%. The statistical error characteristics of the experiment allowed us to obtain an adequate mathematical model of the second order, which confirms that the Fischer table criterion (FT = 2.1) was greater than its calculated value (FR = 1.97).

**Author Contributions:** Conceptualization, A.D. and A.A.; methodology, A.S. and M.G.; software, N.S. and M.M.; validation, A.D.; investigation, A.A.; resources, M.G.; writing—original draft preparation, N.S.; writing—review and editing, A.S.; project administration, A.A.; funding acquisition, A.D. All authors have read and agreed to the published version of the manuscript.

**Funding:** This work was supported by a grant of the Ministry of Science and Higher Education of the Russian Federation for large scientific projects in priority areas of scientific and technological development (grant number 075-15-2020-774).

**Institutional Review Board Statement:** Not applicable.

**Informed Consent Statement:** Not applicable.

**Data Availability Statement:** The raw data supporting the conclusions of this article will be made available by the authors, without undue reservation.

**Conflicts of Interest:** The authors declare that they have no known competing financial interests or personal relationships that could have appeared to influence the work reported in this paper.

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
