# Peer review of "Results of Laboratory Studies of the Automated Sorting System for Root and Onion Crops"

_agronomy, doi:10.3390/agronomy11061257_

Round 1

Reviewer 1 Report

Round 1:

This paper describes the testing of an automated sorting system for root and onion crops. The current state of potato production, along with some of the major challenges, is described to provide the reader with requisite background. An automated sorting system design is described, followed by a description of the experimental setup for testing the machine on potatoes. Results are examined and discussed. The primary contributions of this paper appear to be confirming the sorting process works and outlining a process for determining the limits of the machine parameters.

The technical aspects of the paper seem valid, however there are significant issues with respect to organization and wording that make the paper difficult to comprehend at times. It is unclear on the novelty or value that this paper brings to the literature. That is not to say it does not exist—it just needs to be more clearly articulated in the paper during the next revision.

At first, it seemed implied that the authors had developed the machine, but then it appears they are only testing it.

There is not sufficient discussion on the vision algorithm used for testing or the analysis process.

Consider revising the abstract, including better descriptions of key issues and why it was important to address them, the developed approach and a brief statement about results achieved. The authors note “it is necessary” to do various factors, but it is unclear which are already being done versus which they are newly implementing.

In the introduction, the authors consider a lot of useful papers, stating what was done, how and the results achieved. However, the introduction jumps around topics and is very difficult to follow. There seems to be a lot of good points, but no obvious structure. It is difficult to discern where the review ends and the description of this work begins. The authors need to clearly state the objective of the research and how that has been achieved in this study.

The authors should consider separating the Materials and Methods section into parts and carefully describe the (i) materials selection and reasoning, the (ii) methods and reasoning.

The need for the separate discussion section is not apparent. Discussion should be focused on the results achieved in comparison with previous studies, the alignment with earlier results achieved, or gaps filled by the new findings. None of these seem to be present. Instead, it appears to read like an additional literature review.

There is no conclusion section, and conclusions are not clear from the discussion or data analyzed.  

Line 9: The authors do not mention the name of the Country. Also, reword the opening statement for clarity. Consider: The most common roller and sieve machines for post-harvest processing of root and tuber crops and onions in XYZ (Country name) have a number of disadvantages…

Line 18 – 20: I recommend adding a summary of the results, perhaps a measure of accuracy or damage mitigation, obtained from the sorting system.

Line 27: The phrase “separate potatoes” is not clear. Sorting or merely isolating them? Consider elaborating.

Lines 34 to 48: Seems disjointed from the story line. Consider rewriting. Also, the different examples given seems to be stand alone and does not show implications to the research under consideration. Authors should consider rewriting and find ways to link the different innovations mentioned to the story line -especially the implications, gaps, the knowledge add.

Lines 49 to 99: Too many disjointed and seemingly random paragraphs. Consider creating some structure for the introduction literature review.

Lines 100 to 110: Authors should focus on stating how the research addresses or fill the various gaps identified in the earlier paragraphs. It should also state a brief summary of the work done and the approach and state how it is “superior” to the old methods.

Line 125: It is interesting that a relatively available webcam provides sufficient performance. Could the authors elaborate on the specification analysis that went into the selection?

Lines 195- 248 : Authors should separate this into steps i.e. A graphical flow chart might be best. Each step should be clearly defined, and the output should be clearly laid out.

Figure 3 and 4. It notes these are “a machine” and “a table.” Are these the machines used for the results or no?

Consider including some sort of functional decomposition/flow for the automated sorting device.

Line 275: Reason for choosing v2 is not stated.

Line 276: The variable for Fisher criteria is missing the subscript compared to earlier lines.  

Line 277: Reference might be needed to show why the value qualifies as adequate.

Lines 278 to 292: Consider rewriting. Not clear

Consider explaining the implication of Figure 7 to the results. Also, a description of the color bar variable and units is needed.

The first and last paragraphs in the results section are not clear.

Round 2:

This revision represents significant improvement in the article quality, and the responses and modifications satisfy my prior review commentary. The technical content and technical merit are now clearer, and I recommend the article be accepted with appropriate text editing.

I will restate my prior recommendation that the need for the separate Discussion section is not apparent in the current form. The Results section seems to have much discussion already, whereas the Discussion section appears to read like an additional literature review. I recommend merging the sections for flow. Lastly, the addition of a Conclusion section would be very worthwhile.

Author Response

Round 1:

Здравствуйте уважаемый обозреватель! Отправляю вам ответ на указанные комментарии к статье. "Cm. The attachment"

Round 2:

Здравствуйте уважаемый обозреватель! Отправляю вам ответ на указанные комментарии к статье.

Пункт № 1: Повторяю свою предыдущую рекомендацию о том, что необходимость в отдельном разделе для обсуждения не очевидна в его нынешнем виде. В разделе «Результаты», кажется, уже ведется много дискуссий, а в разделе «Обсуждение» читается как дополнительный обзор литературы. Рекомендую объединять разделы для потока.

Отзыв №1 : В статье разделы «Результаты» и «Обсуждение» объединены в единый раздел «Результаты и обсуждение».

Пункт № 2 Наконец, было бы очень полезно добавить раздел «Заключение».

Отзыв № 2: В статью добавлен раздел «Заключение» ».

Пункт № 3: Требуются умеренные изменения английского языка.

Ответ № 3: Рукопись проверена коллегой-носителем английского языка.

Reviewer 2 Report

Round 1:

Dear, Authors,

Thank You very much for valuable and interesting research. The quality products takes a great place in face of consumers.

Beside it some remarks I'd like to highlight:

  • the Abstract chapter is more for brief description of whole article, so the some part of methodology, results and conclusion have to take a place too.
  • the title of the Fig. 1 describes the parts of the sorting system. Some describe system parts like 5 and 11 missing in a picture and so not clear what are discussing about. If the trays numbered by 9 and 10 are a same, it should be numbered with a one and same number, but if trays functionality are different they require different number and more clear description.
  • Fig. 2 presents block diagram of the root, tubers, and onion sorting line. One part of the block diagram is Microcomputer called Raspberry Pi4. Does it fit more for berries or for onions, tubers and roots sorting too? 
  • More clear explanation of what kind of dimensional characteristics are followed Fig 5? Is it crosscut, circular length or other characteristic are controlled.
  • Paragraph marks has left in each reference numbers on Fig. 6. The numbering have to be remade.
  • The chapters Introduction, Materials and Methods, Results and Discussions numbered with a same number - 1. Check please the article requirements.
  • Article could more clearly disclose the sorting process itself. How the sorting after the camera scaning is performed?
  • How the roots, onions etc are precisely evaluated by cameras during its motion or impurities on a sorting table? 
  • References to tables and figures in the text shown differently, some of them starts with capital letters, so not. Check the article requirements.
  • Text in the paragraph "Declaration of competing interest" have to be corrected.

Round 2:

The article has been moderated and can be accepted in present form.

Author Response

Round 1:

Здравствуйте уважаемый обозреватель! Отправляю вам ответ на комментарии к статье."Cm. The attachment"

Round 2:

Здравствуйте уважаемый обозреватель! Отправляю вам ответ на указанные комментарии к статье.

Пункт № 1: Требуются умеренные изменения английского языка.

Ответ № 1: Рукопись рецензируется коллегой-носителем английского языка.

This manuscript is a resubmission of an earlier submission. The following is a list of the peer review reports and author responses from that submission.

Round 1

Reviewer 1 Report

This paper describes the testing of an automated sorting system for root and onion crops. The current state of potato production, along with some of the major challenges, is described to provide the reader with requisite background. An automated sorting system design is described, followed by a description of the experimental setup for testing the machine on potatoes. Results are examined and discussed. The primary contributions of this paper appear to be confirming the sorting process works and outlining a process for determining the limits of the machine parameters.

The technical aspects of the paper seem valid, however there are significant issues with respect to organization and wording that make the paper difficult to comprehend at times. It is unclear on the novelty or value that this paper brings to the literature. That is not to say it does not exist—it just needs to be more clearly articulated in the paper during the next revision.

At first, it seemed implied that the authors had developed the machine, but then it appears they are only testing it.

There is not sufficient discussion on the vision algorithm used for testing or the analysis process.

Consider revising the abstract, including better descriptions of key issues and why it was important to address them, the developed approach and a brief statement about results achieved. The authors note “it is necessary” to do various factors, but it is unclear which are already being done versus which they are newly implementing.

In the introduction, the authors consider a lot of useful papers, stating what was done, how and the results achieved. However, the introduction jumps around topics and is very difficult to follow. There seems to be a lot of good points, but no obvious structure. It is difficult to discern where the review ends and the description of this work begins. The authors need to clearly state the objective of the research and how that has been achieved in this study.

The authors should consider separating the Materials and Methods section into parts and carefully describe the (i) materials selection and reasoning, the (ii) methods and reasoning.

The need for the separate discussion section is not apparent. Discussion should be focused on the results achieved in comparison with previous studies, the alignment with earlier results achieved, or gaps filled by the new findings. None of these seem to be present. Instead, it appears to read like an additional literature review.

There is no conclusion section, and conclusions are not clear from the discussion or data analyzed.  

Line 9: The authors do not mention the name of the Country. Also, reword the opening statement for clarity. Consider: The most common roller and sieve machines for post-harvest processing of root and tuber crops and onions in XYZ (Country name) have a number of disadvantages…

Line 18 – 20: I recommend adding a summary of the results, perhaps a measure of accuracy or damage mitigation, obtained from the sorting system.

Line 27: The phrase “separate potatoes” is not clear. Sorting or merely isolating them? Consider elaborating.

Lines 34 to 48: Seems disjointed from the story line. Consider rewriting. Also, the different examples given seems to be stand alone and does not show implications to the research under consideration. Authors should consider rewriting and find ways to link the different innovations mentioned to the story line -especially the implications, gaps, the knowledge add.

Lines 49 to 99: Too many disjointed and seemingly random paragraphs. Consider creating some structure for the introduction literature review.

Lines 100 to 110: Authors should focus on stating how the research addresses or fill the various gaps identified in the earlier paragraphs. It should also state a brief summary of the work done and the approach and state how it is “superior” to the old methods.

Line 125: It is interesting that a relatively available webcam provides sufficient performance. Could the authors elaborate on the specification analysis that went into the selection?

Lines 195- 248 : Authors should separate this into steps i.e. A graphical flow chart might be best. Each step should be clearly defined, and the output should be clearly laid out.

Figure 3 and 4. It notes these are “a machine” and “a table.” Are these the machines used for the results or no?

Consider including some sort of functional decomposition/flow for the automated sorting device.

Line 275: Reason for choosing v2 is not stated.

Line 276: The variable for Fisher criteria is missing the subscript compared to earlier lines.  

Line 277: Reference might be needed to show why the value qualifies as adequate.

Lines 278 to 292: Consider rewriting. Not clear

Consider explaining the implication of Figure 7 to the results. Also, a description of the color bar variable and units is needed.

The first and last paragraphs in the results section are not clear.

Author Response

Здравствуйте уважаемый обозреватель! Отправляю вам ответ на указанные комментарии к статье. "Cm. The attachment"

Reviewer 2 Report

Dear, Authors,

Thank You very much for valuable and interesting research. The quality products takes a great place in face of consumers.

Beside it some remarks I'd like to highlight:

  • the Abstract chapter is more for brief description of whole article, so the some part of methodology, results and conclusion have to take a place too.
  • the title of the Fig. 1 describes the parts of the sorting system. Some describe system parts like 5 and 11 missing in a picture and so not clear what are discussing about. If the trays numbered by 9 and 10 are a same, it should be numbered with a one and same number, but if trays functionality are different they require different number and more clear description.
  • Fig. 2 presents block diagram of the root, tubers, and onion sorting line. One part of the block diagram is Microcomputer called Raspberry Pi4. Does it fit more for berries or for onions, tubers and roots sorting too? 
  • More clear explanation of what kind of dimensional characteristics are followed Fig 5? Is it crosscut, circular length or other characteristic are controlled.
  • Paragraph marks has left in each reference numbers on Fig. 6. The numbering have to be remade.
  • The chapters Introduction, Materials and Methods, Results and Discussions numbered with a same number - 1. Check please the article requirements.
  • Article could more clearly disclose the sorting process itself. How the sorting after the camera scaning is performed?
  • How the roots, onions etc are precisely evaluated by cameras during its motion or impurities on a sorting table? 
  • References to tables and figures in the text shown differently, some of them starts with capital letters, so not. Check the article requirements.
  • Text in the paragraph "Declaration of competing interest" have to be corrected.

Author Response

Здравствуйте уважаемый обозреватель! Отправляю вам ответ на комментарии к статье."Cm. The attachment"
